# Geomagnetic Conjugate Observations of Ionospheric Disturbances in response to North Korea Underground Nuclear Explosion on 3 September 2017

Yi Liu, Chen Zhou[*], Qiong Tang, Guanyi Chen, and Zhengyu Zhao

Department of Space Physics, School of Electronic Information, Wuhan University, Wuhan, China

Corresponding to: chenzhou@whu.edu.cn

**Key points:**

1. Geomagnetic conjugate ionospheric disturbances related to UNE were observed by IGS stations and Swarm satellite.

2. Radial propagation velocity from the UNE epicenter was calculated from temporal and spatial distribution of conjugate ionospheric disturbances.

3. The ionospheric disturbances present the evidence of the LAIC electric field penetration process.

## Abstract

We report observations of ionospheric disturbances in response to North Korea underground nuclear explosion (UNE) on 3 September 2017. By using data from IGS (International GNSS Service) stations and Swarm satellite, geomagnetic conjugate ionospheric disturbances were observed. The observational evidences showed that UNE-generated ionospheric disturbances propagated radially from the UNE epicenter with the velocity of ~ 280 m/s. We propose that the ionospheric disturbances are results of electrodynamic process caused by LAIC (Lithosphere-Atmosphere-Ionosphere Coupling) electric field penetration. LAIC electric field can also be mapped to the conjugate hemispheres along the magnetic field line and consequently cause ionospheric disturbances in conjugate regions. The UNE-generated LAIC electric field penetration plays an important role in the ionospheric disturbances in the region of the nuclear test site nearby and the corresponding geomagnetic conjugate points.

**Key words:** geomagnetic conjugate ionospheric disturbances; electrodynamic process; LAIC electric field penetration

## 1 Introduction

Ionospheric disturbances can be generated by various natural processes such as geomagnetic storms, internal electrodynamic instabilities and so forth. Furthermore, human activity can also cause evident ionospheric disturbances. Although underground nuclear explosion (UNE) is detonated deep in the lithosphere, ionospheric disturbances related to UNE can also be observed. By using GNSS-TEC observations, *Park et al.* (2011) reported that traveling ionospheric disturbances (TIDs) with phase velocity of ~273 m/s were generated by UNE in the 25 May 2009 North Korea UNE test. They proposed that acoustic gravity waves (AGWs) generated by the UNE can propagate to ionosphere and cause wavelike disturbances.

While the observations of UNE related ionospheric disturbances have been discussed in (*Park et al.*, 2011; 2013), further investigation is still required to understand the mechanism(s) of ionospheric disturbance generation. Lithosphere-atmosphere-ionosphere coupling (LAIC) mechanisms originally proposed to interpret the linkage between ionospheric disturbances and earthquake activities are the most likely explanation for the ionospheric disturbances in response to UNE. The AGWs theory is one part of LAIC mechanisms (*Liu et al.*, 2016; *Maruyama et al.*, 2016). AGWs excited by the unusual events in lithosphere such as an earthquake or an UNE can propagate to ionospheric height and generate TID and electromagnetic disturbances (*Gokhberg et al.*, 1990; *Pokhotelov et al.*, 1994, 1995, 1999; *Mikhailov et al.*, 2000; *Huang et al.*, 2011; *Jonah et al.*, 2017). However, the AGWs mechanism cannot fully explain all the

observations related to earthquakes. The electrostatic coupling is another candidate for
LAIC mechanisms. During earthquakes, LAIC electric filed or current can be excited
by complex physical and chemical reactions induced by rock rupture and penetrate the
ionosphere to promote plasma disturbances by $E \times B$ motion (*Xu et al.*, 2011; *Zhao &*
*Hao*, 2015). *Zhou et al.* (2017) developed an electric field penetration model for LAIC
and their simulation results showed that the penetration height of LAIC electric field
can reach to 400 km in mid-latitude regions. Because of high electric conductivity along
the geomagnetic field lines, LAIC electric field can also be mapped along geomagnetic
field lines and cause ionospheric disturbances at the geomagnetic conjugate points
(*Ruzhin et al.*, 1998; *Zhang et al., 2009*; *Li & Parrot*, 2017).

In this study, we have used magnetic conjugate GNSS observations and Swarm satellite
to investigate the LAIC electric penetration effects of North Korea UNE on 3
September 2017.

**2 Instrument and Data**
The IGS stations used in this study are located in East Asia and Australia. The
geographical positions of the UNE and the IGS stations are showed in Figure 1. In order
to eliminate the noise and multipath effects of GPS signals, only carrier phase
observations are utilized to derive the relative slant total electron content (STEC). The
time resolution is about 30 s. The ionospheric pierce points (IPPs) height in this study
is assumed at 350 km. Figure 2 shows an example of time series of relative STEC
obtained by SUWN using satellite PRN 28 between 03:00-05:00 UT on 3 September
2017. To calculate the ionospheric disturbances related to UNE from GNSS
observations, the main trends of relative STEC strongly influenced by the Sun's diurnal
cycle need to be removed. In this study, the numerical third-order horizontal 3-point
derivatives of relative STEC are used for extracting the ionospheric disturbances (*Park*
*et al.*, 2011). In the first step, the numerical first-order horizontal 3-point derivatives
are taken as follows:
$$\delta s_i = s_i - \frac{\left(s_{i-1} + s_{i+1}\right)}{2} \qquad i=\{2,...,n\text{-}1\} \qquad (1)$$
where $s_i$ is the $i^{\text{th}}$ data point, $\delta s_i$ is the first derivative, and n is the number of relative
STEC observations. The main relative STEC trends are removed through this process.
Figure 3(a) shows the time series of first-order derivatives of relative STEC. Waves
with small amplitudes occurred at around 3.9 and 4.1 hours, even though it was not
certain whether they were meaningful signals or just noises. The numerical derivative
formula is repeatedly performed on relative STEC derivatives to extract the ionospheric
disturbances related to UNE. The second-order derivatives can be written in the
following expression:
$$\delta\delta s_i = \delta s_i - \frac{\left(\delta s_{i-1} + \delta s_{i+1}\right)}{2} \qquad i=\{2,...,m\text{-}1\} \qquad (2)$$
where $\delta\delta s_i$ is the second derivative, and m is the number of first derivative
observations. Figure 3(b) shows the time series of second-order derivatives of relative
STEC. Compared to the first-order derivatives presented in Figure 3(a), the amplitude
around the 3.9 hour was amplified while others were not significant. The third-order
derivatives are given as follows:
$$\delta\delta\delta s_i = \delta\delta s_i - \frac{\left(\delta\delta s_{i-1} + \delta\delta s_{i+1}\right)}{2} \quad i=\{2,...,l-1\} \tag{3}$$
where $\delta\delta\delta s_i$ is the third derivative, and $l$ is the number of second derivative
observations. Figure 3(c) shows the time series of third-order derivatives of relative
STEC. Compared to the second-order derivatives presented in Figure 3(b), the
disturbances around the 3.9 hour was further amplified. Therefore, compared to the
standard first derivatives, the numerical third-order horizontal –point derivatives can
emphasized the more significant wave components with small amplitudes. Moreover,
to further remove the background noises of third-order derivatives of relative STEC,
the harr wavelet decomposition process is applied to the third-order derivatives.
Equations (4) and (5) give the harr wavelet function and scale function, respectively.
$$\psi_H\left(t\right) = \begin{cases} 1 & 0 \leq t \leq 1/2 \\ -1 & 1/2 \leq t < 1 \\ 0 & \text{others} \end{cases} \tag{4}$$
$$\phi_H\left(t\right) = \begin{cases} 1 & 0 \leq t < 1 \\ 0 & \text{others} \end{cases} \tag{5}$$
Figure 3(d) shows the wavelet de-noised third-order derivatives. From Figure 3(d), it
was found that the background noises in Figure 3(c) were completely removed and only
valuable wave components were retained.

Swarm mission operated by the European Space Agency (ESA) mainly focuses on the
survey of global geomagnetic field and its temporal evolution. Swarm mission consists
of three satellites named Alpha (A), Bravo (B), and Charlie (C). By using the magnetic
field data detected by Vector Field Magnetometer (VFM) on Swarm, the ionospheric
radial current (IRC) density could be calculated by using spatial gradient of residual
magnetic field data through Ampère's law (*Ritter et al.*, 2013). The field-aligned current
(FAC) density could be also obtained by the ratio of the IRC density to the sine of the
magnetic inclination angle. The FAC density and IRC density used in the study were
provided by Swarm level 2 dataset with a time resolution of 1 s. The ionospheric current
disturbances associated with UNE can also be calculated by the above method.

## 3 Observations

**3 Observations**
According to the measurements of China Earthquake Network Center (CENC), the
approximate location of UNE on 3 September, 2017 is at 41.35 °N and 129.11 °E. The
explosive time was at 03:30:01 UTC. The geomagnetic $K$p index was less than 3 and
AE index was less than 500 nT before and after the UNE, which indicates that the
geomagnetic activity was not so active.

Figure 4 shows the time sequences of 3rd-order derivatives of carrier phase derived
relative STEC by GNSS observations from different IGS stations in East Asia and
Australia on 3 September 2017. All the GNSS observations from northern and southern
hemisphere showed obvious short-period fluctuations within 2 hours after the UNE. It
was also found that time delay after the UNE was different according to different IPPs
of GPS signals. Figure 5 presents the IPPs tracks of relative STEC derivatives. In order
to investigate the propagation velocity of ionospheric disturbances, we assumed that
the UNE-generated ionospheric disturbances propagate radially with a certain velocity.

Figure 6 illustrates the satellite Swarm B ionospheric current derivatives. Compared to
observed results of ionospheric current in quiet time, it was seen that the FAC
derivatives and IRC derivatives at conjugate hemispheres both showed obvious short-
period fluctuations after the UNE. The ionospheric current disturbances could reach 0.5
$\mu A \cdot m^{-2} \cdot s^{-3}$.

Based on the UNE-IPPs horizontal distances and the ionospheric disturbances arrival
time, the horizontal propagation velocity of ionospheric disturbances could be
estimated by linear fitting model. The horizontal distance from IPPs to epicenter and
time delay of the UNE-generated ionospheric disturbances (STEC disturbances and
ionospheric current disturbances) are presented in Figure 7. Black triangle and green
triangle presented in Figure 7 represent the position of ionospheric current disturbances
in the northern hemisphere and the geomagnetic conjugate position of ionospheric
current disturbances in the southern hemisphere, respectively. The value of horizontal
velocity obtained by the least square estimation was ~280 m/s.

**4 Discussion**
By utilizing geomagnetic conjugate GNSS TEC observations and ionospheric current
products from Swarm, we introduced the ionospheric disturbances which are
considered as a result of the UNE carried out by North Korea on 3 September 2017.
The method of the numerical third-order horizontal 3-point derivatives was applied to
the GNSS TEC and the ionospheric current of Swarm to extract the ionospheric
disturbances, which can also be found in *Park et al.*, (2011). Ionospheric disturbances
derived from GNSS TEC observations in our study are consistent with the results of
North Korea UNE on 25 May 2009 obtained by *Park et al.* (2011).

The effects of UNE on the ionosphere could be very similar to that of earthquakes on
the ionosphere. In previous studies, AGWs are considered as the most likely mechanism
for atmospheric and ionospheric disturbances excited by UNE or earthquakes
(*Mikhailov et al.*, 2000; *Che et al.*, 2009; *Garrison et al.*, 2010; *Park et al.*, 2011, 2013;
*Yang et al.*, 2012; *Maruyama et al.*, 2016). *Klimenko et al.* (2011) proposed that the
ionospheric disturbances were generated by small-scale internal gravity waves (IGWs)
through propagation and dissipation processes during seismic activity. *Liu et al.* (2016),
and *Chum et al.* (2016, 2018) suggested that co-seismic ionospheric disturbances could
be generated by long-period infrasound waves excited by seismic waves. Based on
GNSS receiver observations over Brazilian sector, *Jonah et al.* (2017) presented
daytime MSTIDs observed in the conjugate hemispheres. They proposed that the
gravity wave-induced polarized electric fields could map into the conjugate hemisphere
and further generate ionospheric disturbances in conjugate region. However, compared
with TEC disturbances induced by MSTIDs presented in *Jonah et al.* (2017),
ionospheric disturbances in response to North Korea UNE in both hemispheres were
smaller and lasted within 5 minutes in our work. Therefore, electric field disturbances
induced by UNE-generated TEC disturbances presented in Figure 4 may be very small
and cannot generate obvious ionospheric disturbances in conjugate region.

Recent researches have shown that earthquake ionospheric disturbances could be
attributed to not only the AGW mechanism but also the electrostatic coupling, which
means the electric field or current penetration into ionosphere induced by earthquakes.
Based on the observations of INTERCOSMOS-BULGARIA-1300 satellite and
DEMETER satellite, *Gousheva et al.* (2008, 2009) and *Zhang et al.* (2014) reported
ionospheric quasi-static electric field perturbations during seismic activities. By using
the magnetometer observations, *Hao et al.* (2013), and *Liu et al.* (2016) showed obvious
ionospheric current and magnetic field perturbations after the Tohoku earthquake. They
proposed that the seismo-traveling atmospheric disturbances (STADs) caused by
infrasonic waves can propagate vertically into the ionosphere and modify the *E* layer
Hall and Pedersen conductivity, resulting in background ionospheric electric field and
magnetic field disturbances. *Pulinets et al.* (2000) proposed a quasi-electrostatic model
for the LAIC mechanism. The simulation results indicated that the abnormal electric
field induced by an earthquake can penetrate into the ionosphere to cause the
ionospheric electric field disturbances (*Sorokin et al.*, 2001). The enhancement of TEC
at the epicenter and its geomagnetic conjugate points were reported by *Liu et al.* (2011),
which indicated that the earthquake-generated electric field penetration can be mapped
along geomagnetic field lines to promote ionospheric disturbances at its conjugate
points by electrodynamic process through $E \times B$ drift. Therefore, the geomagnetic
conjugation effects of ionospheric disturbances in Figure 4 can be explained by the
UNE-generated electric field penetration. A schematic sketch of geomagnetic conjugate
effect related to UNE in the region of the nuclear test site nearby and the corresponding
geomagnetic conjugate region is shown in Figure 8. The UNE-generated electric field
or current penetrates into the ionosphere and further generates an abnormal electric field
at ionospheric altitude. The distribution of ionospheric electric filed showed in Figure
8 were calculated by LAIC electric field penetration model proposed by *Zhou et al.*
(2017). Because of the existence of high conductivity of geomagnetic field, the
abnormal ionospheric electric filed could be mapped along geomagnetic field lines.
Geomagnetic conjugate ionospheric disturbances could be generated by abnormal
ionospheric electric filed through $E \times B$ drift. Our study provides observational
evidences of LAIC electric penetration rather than acoustic gravity wave mechanism.

Geomagnetic conjugate observations in ionosphere have been reported by a few
researchers. *Otsuka et al.* (2002; 2004) reported simultaneous observations of
equatorial airglow depletions and medium-scale TIDs at geomagnetic conjugate points
in both hemispheres by two all-sky imagers. Their results also suggested that
polarization electric field, which is important for airglow depletion and MSTIDs
generation, can be mapped along the field lines.

In our observations, we found that the ionospheric disturbances in both hemispheres
caused by the UNE-generated electric field penetration propagated radially at the
velocity of roughly 280 m/s in Figure 5 and Figure 7. LAIC electric field can be roughly
estimated to be 14.5 mV/m, which is consistent with the magnitude of the earthquake-
generated ionospheric electric field presented by *Zhang et al.* (2014). Figure 6 presents
the results of the ionospheric current disturbances detected by the satellite Swarm B
after the UNE. The reason may be that the ionospheric disturbances from the UNE
propagate here to generate the current disturbances by electrodynamic process.

Moreover, compared with the magnitude and time scale of ionospheric disturbances
caused by earthquakes, there are inconsistencies in our study. Based on IGS station
observations around Tibet and Nepal, *Kong et al.* (2018) reported that TEC disturbances
exceeded 0.3 TECU and lasted for 15-20 minutes during 2015 Nepal earthquake.
However, it was found that the UNE-generated ionospheric disturbances were relatively
smaller and lasted within 5 minutes in Figure 4. The reason for difference of TEC
disturbances may be that earthquake magnitude and background ionosphere are
different.

**5 Summary**
In this study, we have shown that the geomagnetic conjugate observations of GNSS
TEC and ionospheric current from Swarm considered as a response to North Korea
UNE on 3 September 2017. The LAIC electric penetration effects of UNE have been
discussed in details. The main results are summarized as follows:

1. The ionospheric TEC and current disturbances were observed in both hemispheres
after the UNE. According to the spatial-temporal relation, UNE-generated ionospheric
disturbances propagated radially from the explosion epicenter with the velocity of ~
280 m/s.

2. The ionospheric disturbances may be caused by LAIC electric penetration rather than
AGWs. LAIC electric field induced by UNE penetrates into the ionosphere and causes
plasma density disturbances near the nuclear test cite and its conjugate points by
electrodynamic process.

**Acknowledgments**
We thank the use of GPS-TEC data from IGS Data Center of Wuhan University
(http://www.igs.gnsswhu.cn/index.php/Home/DataProduct/igs.html).         We         also
acknowledge the ESA for the Swarm data (https://earth.esa.int/web/guest/swarm/data-
access). The work is supported by the National Natural Science Foundation of China
(NSFC grant No. 41574146 and 41774162).

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

 **Figure Captions**

**Figure 1**. The positions of UNE and IGS stations. The position of 3 September 2017
North Korea UNE is represented by black hollow start mark. The locations of IGS
stations in both hemispheres are represented by red and blue squares, respectively.
Lines of constant geomagnetic latitude are represented by black dashed lines.
**Figure 2.** An example of time series of relative STEC obtained by SUWN using
satellite PRN 28 between 03:00-05:00 UT on 3 September 2017. The explosive time is
represented by the red line.
**Figure 3.** The time sequences of derivatives of relative STEC obtained by SUWN
station using satellite PRN 28 between 03:00-05:00 UT on 3 September 2017. (a) first-
order derivatives, (2) second-order derivatives, (c) third-order derivatives, and (d)
wavelet de-noised third-order derivatives. The explosive time is represented by the red
line.
**Figure 4.** The time sequences of 3-order derivatives of carrier phase derived relative
STEC by GNSS observations from different IGS stations in East Asia (left and middle
column) and Australia (right column) on 3 September 2017. The blue lines indicate the
wavelet de-noised 3-order derivative of relative STEC. The black lines indicate the GPS
signal's elevation between the GNSS satellite and IGS stations. The explosive time is
represented by the red line.
**Figure 5.** The IPPs tracks of relative STEC derivatives. The red lines indicate the IPPs
tracks obtained by IGS stations in the northern hemisphere. The blue lines indicate the
magnetic conjugate positions of the IPPs tracks obtained by IGS stations in the southern
hemisphere. The positions of the maximum amplitudes of relative STEC derivatives in
the northern hemisphere are represented by red triangles. The geomagnetic conjugate
positions of the maximum amplitudes of relative STEC derivatives in the southern
hemisphere are represented by blue triangles.
**Figure 6.** Results of Swarm B ionospheric current data analysis for the 2017 UNE: (a),
(c), and (e) are the FAC, (b), (d), (f) are the IRC. From top to bottom, they indicate
observations of Swarm B on 19 August 2017 (quiet time), 3 September 2017 (UNE
time), and 18 September 2017 (quiet time), respectively. The ionospheric current
disturbances in response to UNE are represented by the red rectangles.
**Figure 7.** Horizontal distance-time data for the UNE-generated ionospheric
disturbances. The black line indicates the fitting curve obtained by the least square
method. The gray lines represent the boundaries of 95% confidence intervals. The red
and blue triangles indicate same meanings as in Figure 5. The black triangle represents
the position of ionospheric current disturbances in the northern hemisphere. The green
triangle represents the geomagnetic conjugate position of ionospheric current
disturbances in the southern hemisphere.
**Figure 8.** A sketch of geomagnetic conjugate effect related to UNE in the region of the
nuclear test site nearby and the corresponding geomagnetic conjugate region.

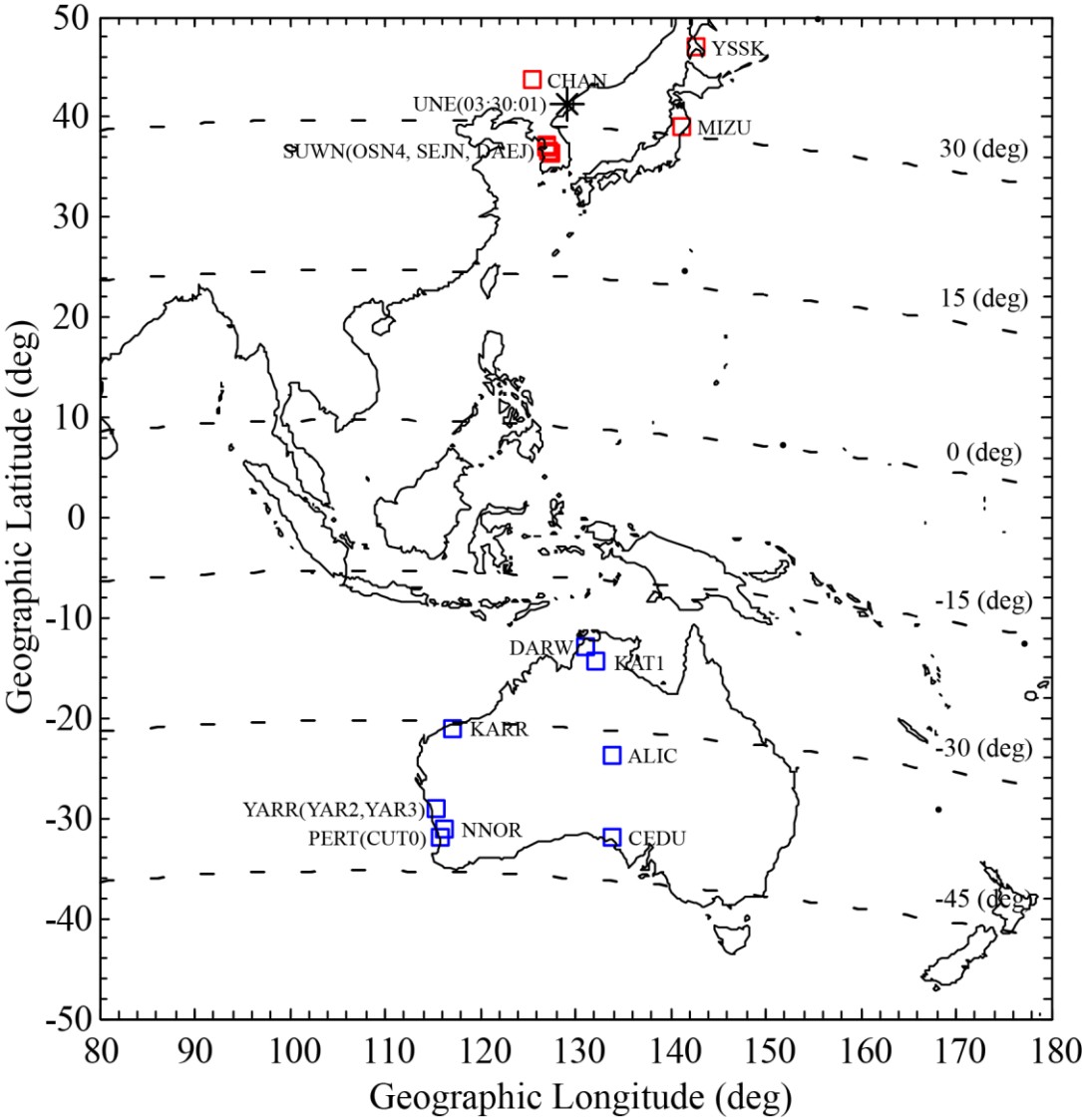

**Figure 1.** The positions of UNE and IGS stations. The position of 3 September 2017 North Korea UNE is represented by black hollow start mark. The locations of IGS stations in both hemisphere are represented by red and blue squares, respectively. Lines of constant geomagnetic latitude are represented by black dashed lines.

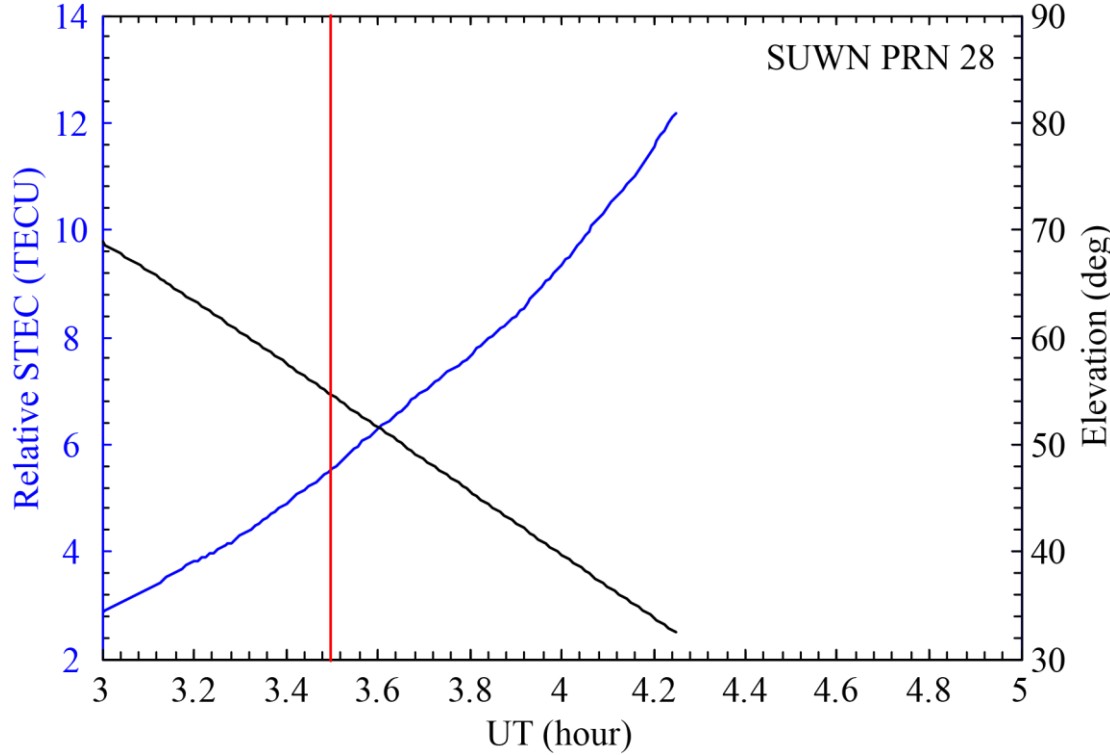


**Figure 2.** An example of time series of relative STEC obtained by SUWN using
satellite PRN 28 between 03:00-05:00 UT on 3 September 2017. The explosive time
is represented by the red line.

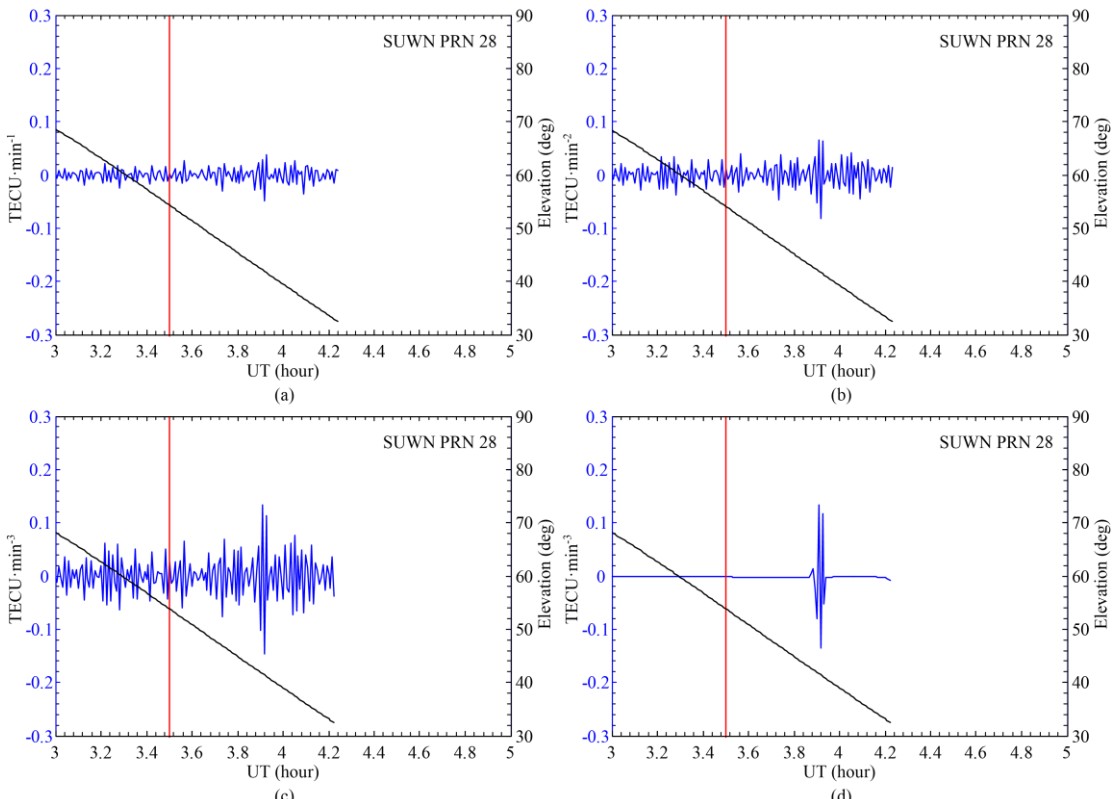


**Figure 3.** The time sequences of derivatives of relative STEC obtained by SUWN
station using satellite PRN 28 between 03:00-05:00 UT on 3 September 2017. (a)
first-order derivatives, (2) second-order derivatives, (c) third-order derivatives, and
(d) wavelet de-noised third-order derivatives. The explosive time is represented by the
red line.

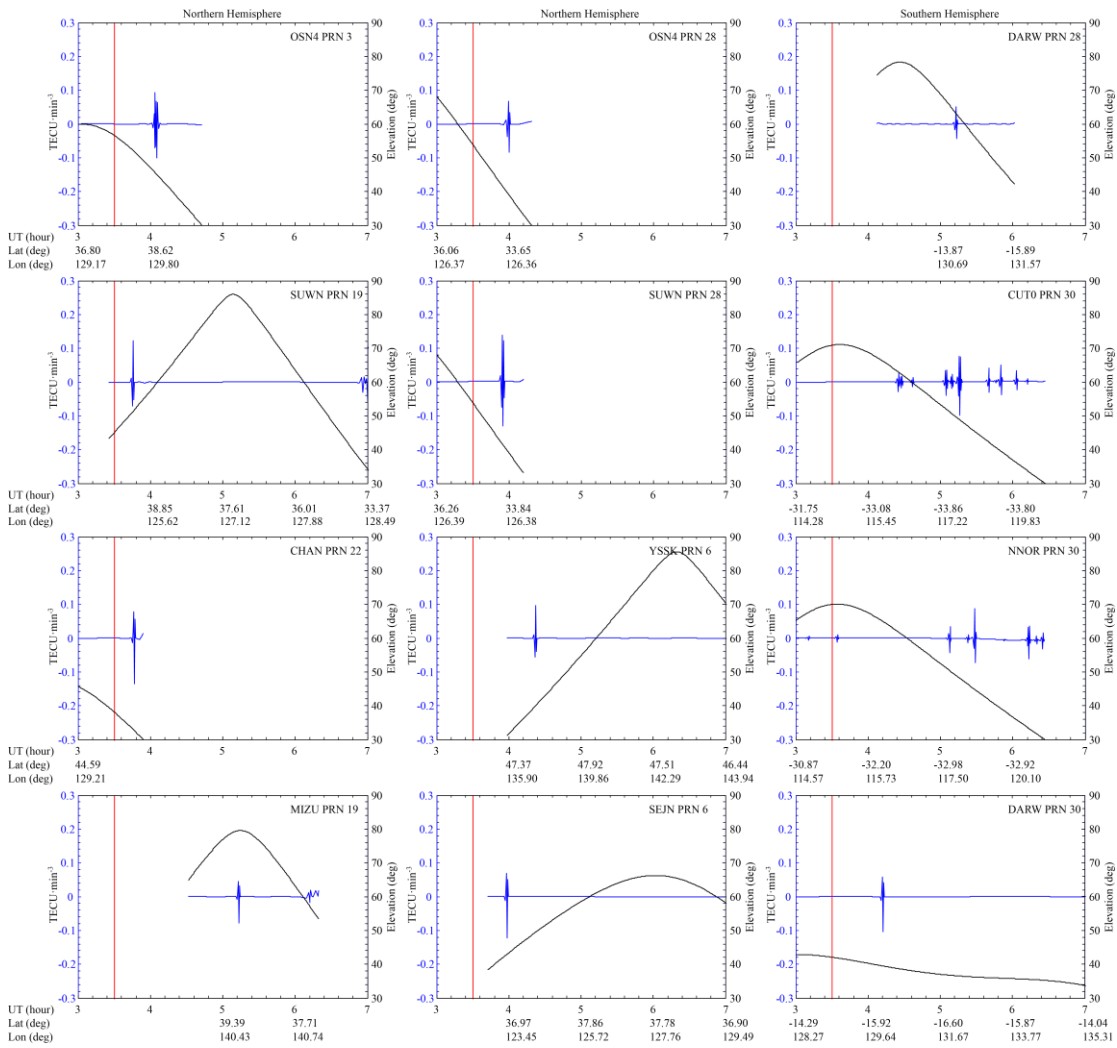


**Figure 4.** The time sequences of 3-order derivatives of carrier phase derived relative STEC by GNSS observations from different IGS stations in East Asia (left and middle column) and Australia (right column) on 3 September 2017. The blue lines indicate the wavelet de-noised 3-order derivative of relative STEC. The black lines indicate the GPS signal's elevation angle between the GNSS satellite and IGS stations. The explosive time is represented by the red line.

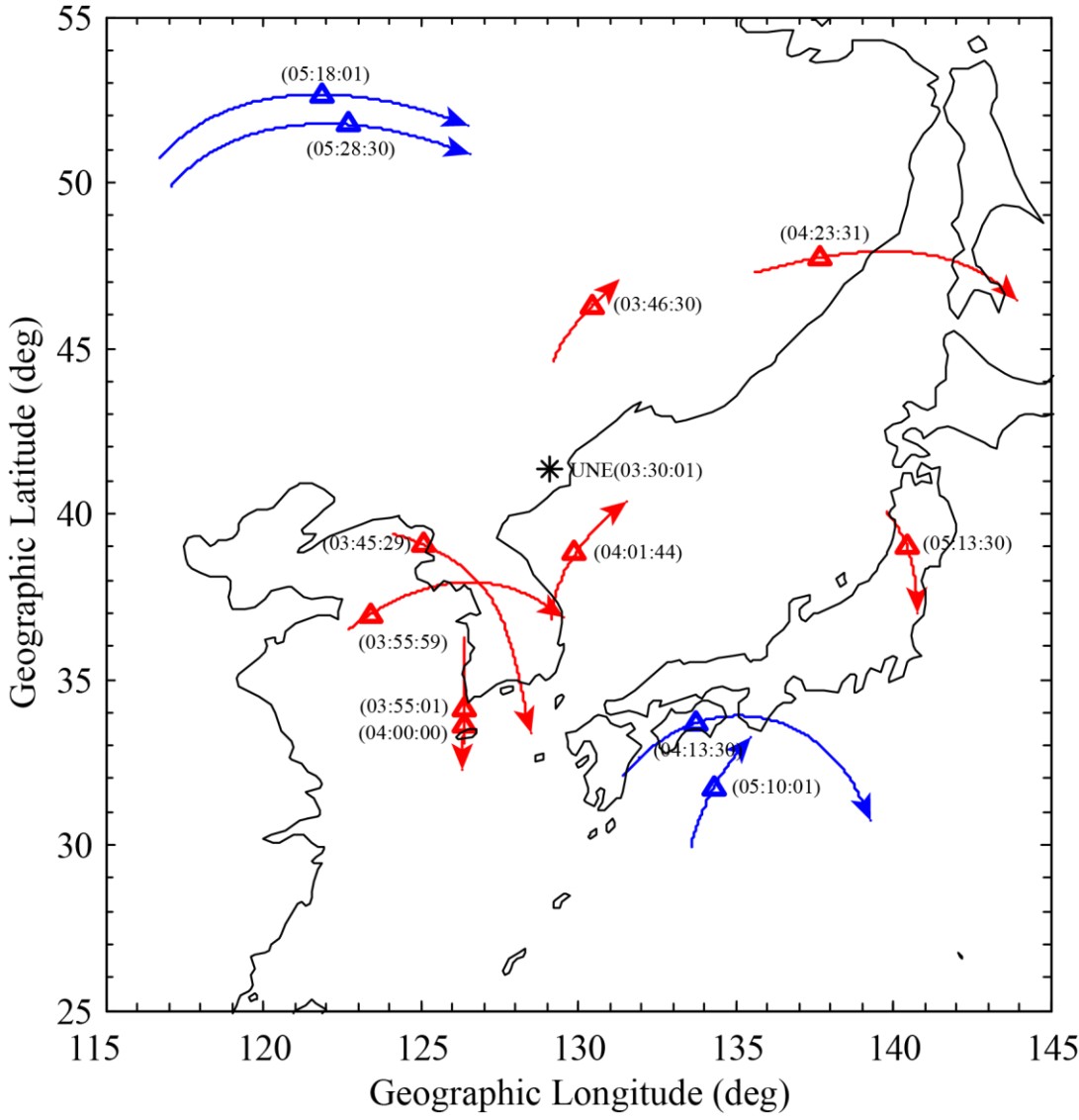


**Figure 5.** The IPPs tracks of relative STEC derivatives. The red lines indicate the
IPPs tracks obtained by IGS stations in the northern hemisphere. The blue lines
indicate the magnetic conjugate positions of the IPPs tracks obtained by IGS stations
in the southern hemisphere. The positions of the maximum amplitudes of relative
STEC derivatives in the northern hemisphere are represented by red triangles. The
geomagnetic conjugate positions of the maximum amplitudes of relative STEC
derivatives in the southern hemisphere are represented by blue triangles.

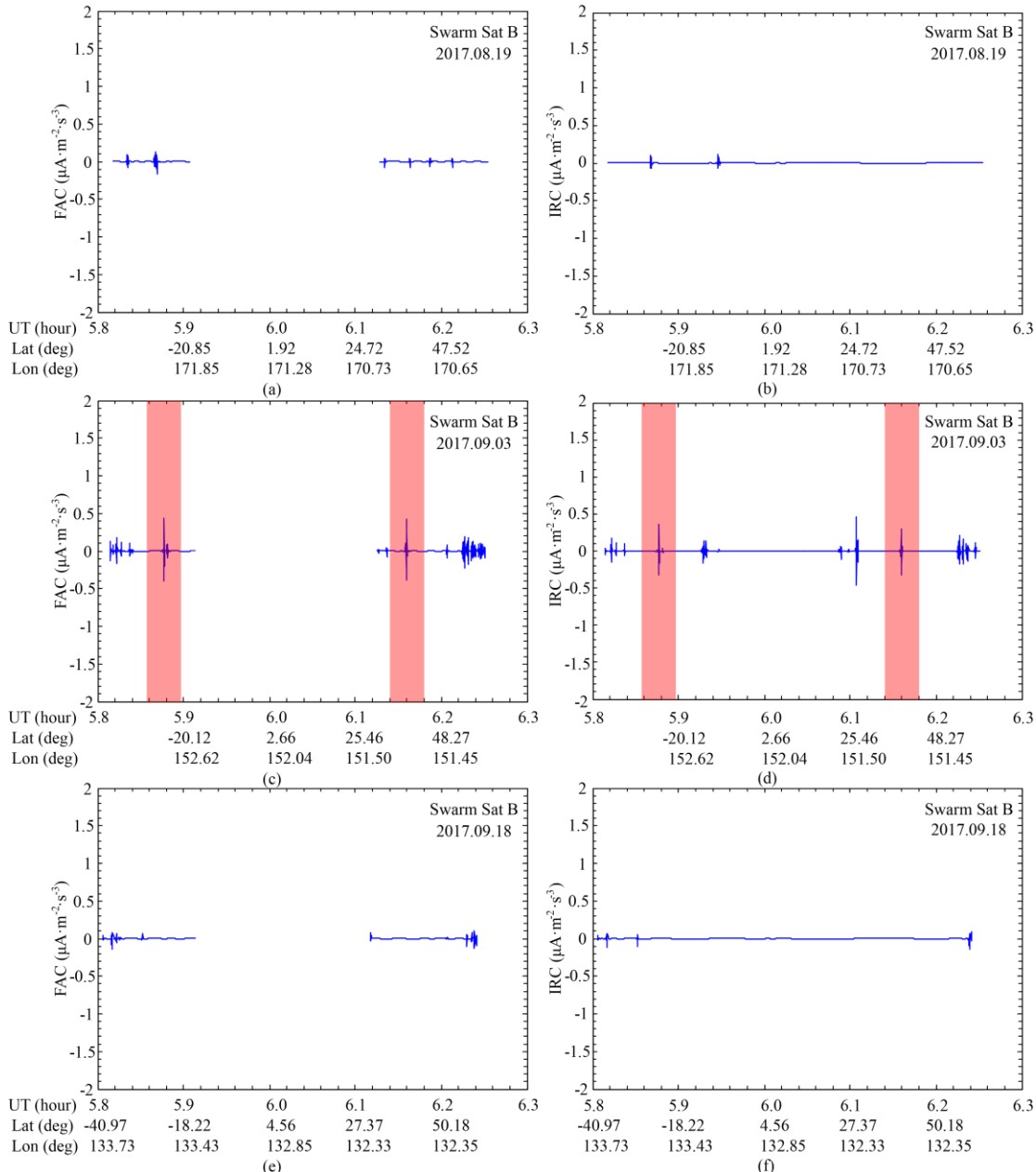

**Figure 6.** Results of Swarm B ionospheric current data analysis for the 2017 UNE: (a), (c), and (e) are the FAC, (b), (d), (f) are the IRC. From top to bottom, they indicate observations of Swarm B on 19 August 2017 (quiet time), 3 September 2017 (UNE time), and 18 September 2017 (quiet time), respectively. The ionospheric current disturbances in response to UNE are represented by the red rectangles.

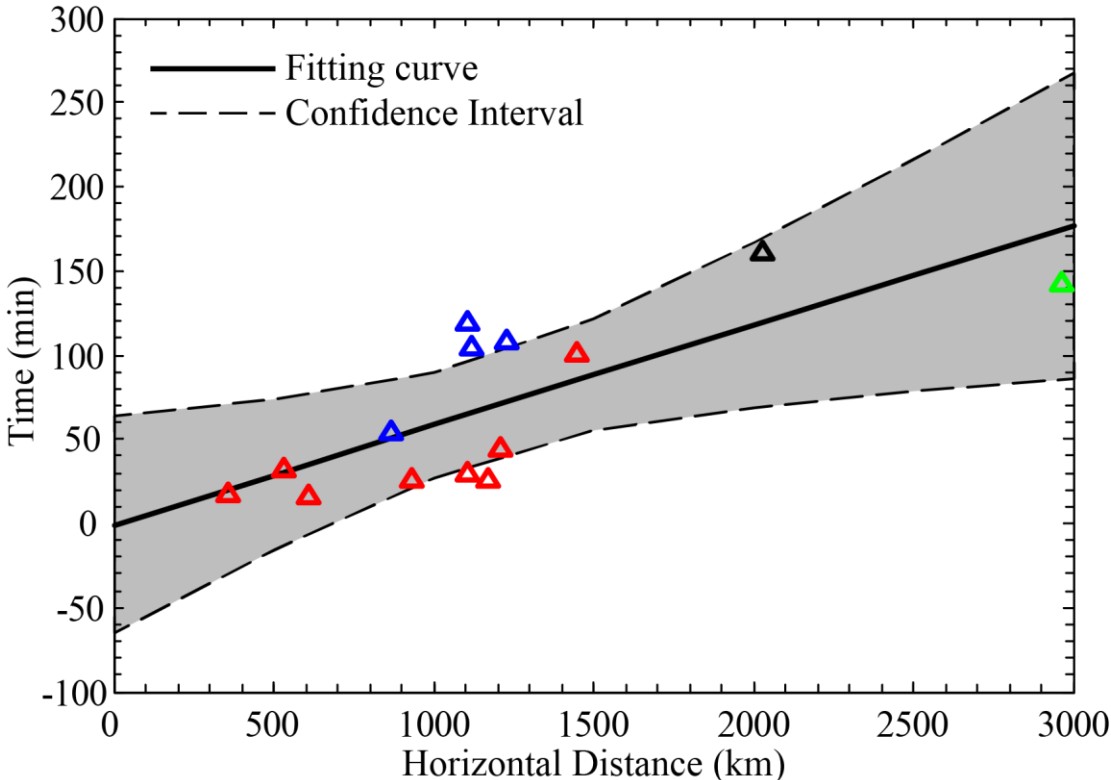

**Figure 7.** Horizontal distance-time data for the UNE-generated ionospheric disturbances. The black line indicates the fitting curve obtained by the least square method. The gray lines represent the boundaries of 95% confidence intervals. The red and blue triangles indicate same meanings as in Figure 5. The black triangle represents the position of ionospheric current disturbances in the northern hemisphere. The green triangle represents the geomagnetic conjugate position of ionospheric current disturbances in the southern hemisphere.

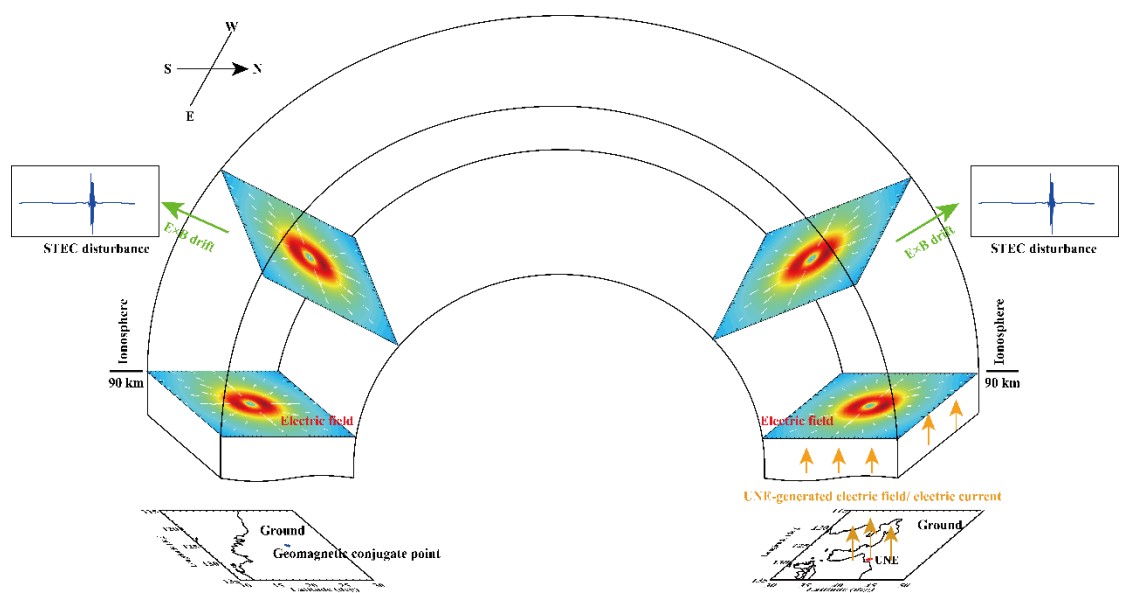

**Figure 8.** A sketch of geomagnetic conjugate effect related to UNE in the region of

the nuclear test site nearby and the corresponding geomagnetic conjugate region.