# Peer review of "Geomagnetic Conjugate Observations of Ionospheric Disturbances in response to North Korea Underground Nuclear Explosion on 3 September 2017"

_Annales Geophysicae, 2018_

## Referee Comment (RC1) · Anonymous Referee #1 · 29 Nov 2018

**Review of the manuscript** entitled "Geomagnetic Conjugate Observations of Ionospheric Disturbances in response to North Korea Underground Nuclear Explosion on 3 September 2017" by Liu et al., submitted for a possible publication in Annales Geophysicae [angeo-2018-122]

**General comment**

The manuscript describes observation of ionospheric disturbances induced by underground nuclear explosion (UNE) in North Korea on 3 September 2017. The ionospheric disturbances were observed both on the northern hemisphere and on southern hemisphere around conjugate point. The manuscript is reasonable well written, the subject is suitable for publication in Annales Geophysicae. I think that several points could be addressed more carefully to improve quality of the paper (see the specific comments). I recommend a moderate revision.

**Specific comments**

a) Section 2, the method of data analysis should be described in more detail. Specifically, the third-order horizontal 3-point derivative should be defined. It should be mentioned why such a derivative was used, and discussed its advantage with respect to standard first derivative. The authors reference to paper by Park et al. (2011) in this respect, however, I have not found a sufficient definition and discussion related to this derivative in their paper. Also, the procedure of removing background noise by using wavelet decomposition should be briefly described.

b)line 113-114 and Figure 3, I suggest comparison with average values calculated for 15 quite days before and after the UNE event rather than for only one day before the event.
Also, I would recommend locating the modified text related to current Figure 3 after the text related to current Figure 5 (after line 125), and renumbering Figure 3 to Figure 6 (renumber Figure 4 to Figure 3). Current Figures 2 and 4 and the corresponding texts are closely related. I thing that the flow of information will be more logical in the suggested re-organization. In addition, insert explanation of black and green triangles in the text related to Figure 5 (current lines 123-125).

c) Discussion, paragraph related to similarity with earthquakes. It should be mentioned, e.g., after the sentence Klimenko et al (2011)…that there were several studies that showed that co-seismic ionospheric disturbances were caused by long-period infrasound waves that propagated nearly vertically to ionospheric heights (Chum et al., 2016; Liu et al., 2016, Chum et al., 2018 and references therein).

Chum, J., J.-Y. Liu, K. Podolská, T. Šindelářová (2018), Infrasound in the ionosphere from earthquakes and typhoons, *J. Atmos. Sol. Terr. Phys.*, 171, 72-82, doi:/10.1016/j.jastp.2017.07.022

Chum, J., M. A. Cabrera, Z. Mošna, M. Fagre, J. Baše, and J. Fišer (2016), Nonlinear acoustic waves in the viscous thermosphere and ionosphere above earthquake, *J. Geophys. Res. Space Physics, 121*, doi:10.1002/2016JA023450.

Liu et al., (2016) is already in the references

d) lines 180-181, LAIC electric field can be roughly estimated to be 11 mV/m. Specify the method of estimation.

e)Figure 5, related text and discussion. Specify, if the least square fitting was done under assumption that the fitted line goes through the beginning (point [0; 0]) or if an arbitrary offset along the vertical axis was admitted. If the arbitrary offset (preferred in my opinion) is admitted then from the obtained time delay at distance 0, one could say something about the time delay between explosion and ionospheric perturbation just above the explosion. Likely, one should have observation close to the explosion to obtain reliable results (time delay with sufficient precision). Anyway, theoretically, knowledge of this time delay could help to distinguish if the electric fields penetrated from below (from the ground) or if they were generated in the ionosphere. Note that there is a possibility that mechanic perturbations caused by AGWs change the electric conductivity in the lower ionosphere, which in turn, in the presence of (zonal) electric fields can cause horizontal perturbation of these background electric fields and associated currents that can be detected as geomagnetic perturbations (e.g. Liu et al., 2016). A possibility of such a mechanism should be briefly mentioned/discussed for completeness.

**Technical comments (Minor or formal comments and language suggestions)**
-line 45, naturally processes-> natural processes
-lines 46-47, coupled upper atmospheric variations – specify or remove
-line 71, conductivity of the geomagnetic…->conductivity along the geomagnetic
-line 91, …temporal evolution which consists of…->…temporal evolution. SWARM mission consists of…
-line 153, …indicated the abnormal…->…indicated that the abnormal…
Also, add a suitable reference after this sentence

---

## Referee Comment (RC2) · Anonymous Referee #2 · 18 Dec 2018

This MS reports observations of ionospheric prturbations in responce to North Korea undeground nuclear explosion on September 2017. By using data from the ground and satellite facilities the ionospheric disturbances in the conjugate point have been detected. Similar obsevations were mentioned in the earlier publication by Gokhberg et al. (1990). The authors of the present MS propose that these perturbations are results of electrodynamic process caused by LAIC electric field penetration. The paper is concisely written and contains important results. However, the MS is not free from some defficiencies described below.

Line 62. Replace Gohberg by Gokhberg.

[Figure]

Line 63. Replace Mikhailv by Mikhailov.

Lines 213-215. Please replace the title of the reference to "Acoustic disturbance induced by underground neuclear explosion as a source of electrostatic turbulence in the magnetosphere".

Line 215. Replace P568 to P568-574.

After these corrections the MS can be published in Ann. Geophys.

---

## Referee Comment (RC4) · Anonymous Referee #3 · 21 Dec 2018

Paper by Liu et al. "Geomagnetic. . ." promises to be an interesting and important study. However, in the current form the presentation of observational results is not convincing.

Authors discussed the magnitude of expected electric field disturbance about 11 mV/m (p. 9). How this estimate was obtained? It would be better to discuss the magnitude and waveform of TEC disturbance, that authors had actually measured.

Fig. 1. According to this map, there are several GPS stations in the vicinity of nuclear testing ground. Why not to provide TEC data from both the conjugate point and the same hemisphere site?

Fig. 2. In this plot only the moment of TEC disturbance can be seen. However, the

[Figure]

waveform of TEC disturbance is not shown anywhere. Additional Figure with extended time scale is needed.

Fig. 3. The same problem with this plot. Only the moment of FAC impulse can be seen, but not its waveform. Additional Figure with extended time scale is needed. Plot for another day is not necessary.

Editorial comments: Fig. 1. Lines with geomagnetic coordinates are needed.

The reference to Ren et al. (2012) is absolutely irrelevant.

All the names in ref. at line 213 are misspelled.

Few comments concerning interpretation: Theoretical model of FAC generation at the front of the acoustic pulse has been presented in [Pokhotelov O.A., Parrot M., Pilipenko V.A., Fedorov E.N., Surkov V.V., and Gladyshev V.A., Response of the ionosphere to natural and man-made acoustic sources, Annales Geophysicae, 13, N11, 1197-1210, 1995; Pokhotelov O.A., Pilipenko V.A., Fedorov E.N., Stenflo L., and Shukla P.K., Induced electromagnetic turbulence in the ionosphere and the magnetosphere, Physica Scripta, 50, 600-605, 1994; Pokhotelov, O.A., Pilipenko V.A., and Parrot M., Strong atmospheric disturbances as a possible origin of inner zone particle diffusion, Annales Geophysicae, 17, 526-532, 1999].

---

## Author Comment (AC1) · 28 Dec 2018

Review of the manuscript entitled "Geomagnetic Conjugate Observations of Ionospheric Disturbances in response to North Korea Underground Nuclear Explosion on 3 September 2017" by Liu et al., submitted for a possible publication in Annales Geophysicae [angeo-2018-122] General comment The manuscript describes observation of ionospheric disturbances induced by underground nuclear explosion (UNE) in North Korea on 3 September 2017. The ionospheric disturbances were observed both on the northern hemisphere and on southern hemisphere around conjugate point. The manuscript is reasonable well written, the subject is suitable for publication in Annales

[Figure]

Geophysicae. I think that several points could be addressed more carefully to improve quality of the paper (see the specific comments). I recommend a moderate revision. Specific comments a) Section 2, the method of data analysis should be described in more detail. Specifically, the third-order horizontal 3-point derivative should be defined. It should be mentioned why such a derivative was used, and discussed its advantage with respect to standard first derivative. The authors reference to paper by Park et al. (2011) in this respect, however, I have not found a sufficient definition and discussion related to this derivative in their paper. Also, the procedure of removing background noise by using wavelet decomposition should be briefly described.

Response: We appreciate the reviewer for the valuable comment. The IGS stations used in this study are located in East Asia and Australia. The geographical positions of the UNE and the IGS stations are showed in Figure 1. In order to eliminate the noise and multipath effects of GPS signals, only carrier phase observations are utilized to derive the relative slant total electron content (STEC). The time resolution is about 30 s. The ionospheric pierce points (IPPs) height in this study is assumed at 350 km. Figure 2 shows an example of time series of relative STEC obtained by SUWN using satellite PRN 28 between 03:00-05:00 UT on 3 September 2017. To calculate the ionospheric disturbances related to UNE from GNSS observations, the main trends of relative STEC strongly influenced by the Sun's diurnal cycle need to be removed. In this study, the numerical third-order horizontal 3-point derivatives of relative STEC are used for extracting the ionospheric disturbances (Park et al., 2011). In the first step, the numerical first-order horizontal 3-point derivatives are taken as follows: (1) where is the ith data point, is the first derivative, and n is the number of relative STEC observations. The main relative STEC trends are removed through this process. Figure 3(a) shows the time series of first-order derivatives of relative STEC. Waves with small amplitudes occurred at around 3.9 and 4.1 hours, even though it was not certain whether they were meaningful signals or just noises. The numerical derivative formula is repeatedly performed on relative STEC derivatives to extract the ionospheric disturbances related to UNE. The second-order derivatives can be written in the following expression: (2)

[Figure]

where is the second derivative, and m is the number of first derivative observations. Figure 3(b) shows the time series of second-order derivatives of relative STEC. Compared to the first-order derivatives presented in Figure 3(a), the amplitude around the 3.9 hour was amplified while others were not significant. The third-order derivatives are given as follows: (3) where is the third derivative, and l is the number of second derivative observations. Figure 3(c) shows the time series of third-order derivatives of relative STEC. Compared to the second-order derivatives presented in Figure 3(b), the disturbances around the 3.9 hour was further amplified. Therefore, compared to the standard first derivatives, the numerical third-order horizontal –point derivatives can emphasized the more significant wave components with small amplitudes. Moreover, to further remove the background noises of third-order derivatives of relative STEC, the harr wavelet decomposition process is applied to the third-order derivatives. Equations (4) and (5) give the harr wavelet function and scale function, respectively. (4) (5) Figure 3(d) shows the wavelet de-noised third-order derivatives. From Figure 3(d), it was found that the background noises in Figure 3(c) were completely removed and only valuable wave components were retained.

b)line 113-114 and Figure 3, I suggest comparison with average values calculated for 15 quite days before and after the UNE event rather than for only one day before the event. Also, I would recommend locating the modified text related to current Figure 3 after the text related to current Figure 5 (after line 125), and renumbering Figure 3 to Figure 6 (renumber Figure 4 to Figure 3). Current Figures 2 and 4 and the corresponding texts are closely related. I thing that the flow of information will be more logical in the suggested re-organization. In addition, insert explanation of black and green triangles in the text related to Figure 5 (current lines 123-125).

Response: We appreciate the reviewer for the valuable comment. We present the quiet time FAC derivatives and IRC derivatives for 15 quiet days before and after the UNE event in Figure 6. It was found that ionospheric current derivatives remained smooth in quiet time. By comparing with quiet time observations, obvious short-period fluctuations of ionospheric current derivatives at conjugate hemispheres were observed after the UNE in Figure 6(c) and Figure 6(d).

Figure 7 shows the horizontal distance from IPPs to epicenter and time delay of the UNE-generated ionospheric disturbances (STEC disturbances and ionospheric current disturbances). Therefore, we have renumbered Figure 5 to Figure 6 (renumber Figure 6 to Figure 5). Black triangle and green triangle presented in Figure 7 represent the position of ionospheric current disturbances in the northern hemisphere and the geomagnetic conjugate position of ionospheric current disturbances in the southern hemisphere, respectively.

c) Discussion, paragraph related to similarity with earthquakes. It should be mentioned, e.g., after the sentence Klimenko et al (2011)...that there were several studies that showed that co-seismic ionospheric disturbances were caused by long-period infrasound waves that propagated nearly vertically to ionospheric heights (Chum et al., 2016; Liu et al., 2016, Chum et al., 2018 and references therein). Chum, J., J.-Y. Liu, K. Podolská, T. Šindelářová (2018), Infrasound in the ionosphere from earthquakes and typhoons, J. Atmos. Sol. Terr. Phys., 171, 72-82, doi:/10.1016/j.jastp.2017.07.022 Chum, J., M. A. Cabrera, Z. Mošna, M. Fagre, J. Baše, and J. Fišer (2016), Nonlinear acoustic waves in the viscous thermosphere and ionosphere above earthquake, J. Geophys. Res. Space Physics, 121, doi:10.1002/2016JA023450. Liu et al., (2016) is already in the references

Response: We appreciate the reviewer for the helpful suggestion. We have followed the reviewer's suggestion and added these references in the revised manuscript.

d) lines 180-181, LAIC electric field can be roughly estimated to be 11 mV/m. Specify the method of estimation.

Response: We appreciate the reviewer for the valuable comment. LAIC electric field can be roughly estimated by the following expression: (6) where is the total propagation velocity of ionospheric disturbances, E is LAIC electric field, and B is the magnetic field.

Based on the LAIC electric field penetration model proposed by Zhou et al. (2017), it is found that LAIC electric field is perpendicular to the magnetic field. Therefore, total propagation velocity of ionospheric disturbances generated through E×B drift can be calculated by . The value of horizontal velocity obtained by the least square estimation was ∼280 m/s in this study. Total magnetic field intensity and magnetic inclination angle I around UNE test site calculated by International Geomagnetic Reference Field (IGRF) model were 4.39*10ˆ-5 T and 57.90°, respectively. Therefore, LAIC electric field can be roughly estimated by equation (6) to be 14.5 mV/m.

e)Figure 5, related text and discussion. Specify, if the least square fitting was done under assumption that the fitted line goes through the beginning (point [0; 0]) or if an arbitrary offset along the vertical axis was admitted. If the arbitrary offset (preferred in my opinion) is admitted then from the obtained time delay at distance 0, one could say something about the time delay between explosion and ionospheric perturbation just above the explosion. Likely, one should have observation close to the explosion to obtain reliable results (time delay with sufficient precision). Anyway, theoretically, knowledge of this time delay could help to distinguish if the electric fields penetrated from below (from the ground) or if they were generated in the ionosphere. Note that there is a possibility that mechanic perturbations caused by AGWs change the electric conductivity in the lower ionosphere, which in turn, in the presence of (zonal) electric fields can cause horizontal perturbation of these background electric fields and associated currents that can be detected as geomagnetic perturbations (e.g. Liu et al., 2016). A possibility of such a mechanism should be briefly mentioned/discussed for completeness.

Response: We appreciate the reviewer for the valuable comment. We agree with reviewer's point that the knowledge of this time delay between explosion and ionospheric perturbation just above the explosion could help to distinguish if the electric fields penetrated from below (from the ground) or if they were generated in the ionosphere. However, in this work, we found that there is no relative STEC observations

from IGS stations close to the UNE test site during the UNE events. Therefore, it is no way to investigate the time delay.

The physical mechanism that the electric field perturbations can be generated in the ionosphere has been briefly discussed in the revised manuscript. Please see Page 10 Line 186-192.

Technical comments (Minor or formal comments and language suggestions) -line 45, naturally processes-> natural processes -lines 46-47, coupled upper atmospheric variations – specify or remove -line 71, conductivity of the geomagnetic. . .->conductivity along the geomagnetic -line 91, . . .temporal evolution which consists of. . .->. . .temporal evolution. SWARM mission consists of. . . -line 153, . . .indicated the abnormal. . .->. . .indicated that the abnormal. . . Also, add a suitable reference after this sentence

Response: We appreciate the reviewer for the valuable comment. We have corrected accordingly. We would like to thank the reviewer again for the valuable comments, which help a lot to improve the quality of the present paper. We hope that the reviewers will be satisfied with our responses and revisions, and we look forward to hearing from the reviewers soon.

Reference: Park, J., Frese, R. R. B. von, Grejner‐Brzezinska, D. A., Morton, Y., and Gaya‐Pique, L. R.: Ionospheric detection of the 25 May 2009 North Korean underground nuclear test, Geophys. Res. Lett., 38, L22802, 2011. Zhou, C., Liu, Y., Zhao, S., Liu, J., Zhang, X., Huang, J., Shen, X., Ni, B., and Zhao, Z.: An electric field penetration model for seismo-ionospheric research, Adv. Space Res., 60(10), 2217-2232, 2017.

Please also note the supplement to this comment:
https://www.ann-geophys-discuss.net/angeo-2018-122/angeo-2018-122-AC1-supplement.zip
2018.

ANGEOD

Interactive
comment

**Fig. 1.** Figure 1

SUWN PRN 28

**Fig. 2.** Figure 2

**Fig. 3.** Figure 3

**Fig. 4.** Figure 4

[Figure]

**Fig. 5.** Figure 5

**Fig. 6.** Figure 6

**Fig. 7.** Figure 7

[Figure]

**Fig. 8.** Figure 8

---

## Author Comment (AC2) · 28 Dec 2018

This MS reports observations of ionospheric perturbations in responce to North Korea undeground nuclear explosion on September 2017. By using data from the ground and satellite facilities the ionospheric disturbances in the conjugate point have been detected. Similar obsevations were mentioned in the earlier publication by Gokhberg et al. (1990). The authors of the present MS propose that these perturbations are results of electrodynamic process caused by LAIC electric field penetration. The paper is concisely written and contains important results. However, the MS is not free from some defficiencies described below. Line 62. Replace Gohberg by Gokhberg. Line 63.

[Figure]

Replace Mikhailv by Mikhailov. Lines 213-215. Please replace the title of the reference to "Acoustic disturbance induced by underground neuclear explosion as a source of electrostatic turbulence in the magnetosphere". Line 215. Replace P568 to P568-574

Response: We appreciate the reviewer for the valuable comment. We have corrected accordingly. We would like to thank the reviewer again for the valuable comments, which help a lot to improve the quality of the present paper. We hope that the reviewers will be satisfied with our responses and revisions, and we look forward to hearing from the reviewers soon.

Please also note the supplement to this comment:
https://www.ann-geophys-discuss.net/angeo-2018-122/angeo-2018-122-AC2-supplement.zip

---

## Author Comment (AC4) · 28 Dec 2018

Paper by Liu et al. "Geomagnetic. . ." promises to be an interesting and important study. However, in the current form the presentation of observational results is not convincing. Authors discussed the magnitude of expected electric field disturbance about 11 mV/m (p. 9). How this estimate was obtained? It would be better to discuss the magnitude and waveform of TEC disturbance, that authors had actually measured.

Response: We appreciate the reviewer for the valuable comment. LAIC electric field can be roughly estimated by the following expression: (6) where is the total propagation velocity of ionospheric disturbances, E is LAIC electric field, and B is the magnetic field.

Based on the LAIC electric field penetration model proposed by Zhou et al. (2017), it is found that LAIC electric field is perpendicular to the magnetic field. Therefore, total propagation velocity of ionospheric disturbances generated through E×B drift can be calculated by . The value of horizontal velocity obtained by the least square estimation was ∼280 m/s in this study. Total magnetic field intensity and magnetic inclination angle I around UNE test site calculated by International Geomagnetic Reference Field (IGRF) model were 4.39*10ˆ-5 T and 57.90°, respectively. Therefore, LAIC electric field can be roughly estimated by equation (6) to be 14.5 mV/m.

Compared with the magnitude and time scale of ionospheric disturbances caused by earthquakes, there are inconsistencies in our study. Based on IGS station observations around Tibet and Nepal, Kong et al. (2018) reported that TEC disturbances exceeded 0.3 TECU and lasted for 15-20 minutes during 2015 Nepal earthquake. However, it was found that the UNE-generated ionospheric disturbances were relatively smaller and lasted within 5 minutes in Figure 4. Therefore, it is possible to distinguish natural earthquakes and UNE events based on GNSS observations.

Fig. 1. According to this map, there are several GPS stations in the vicinity of nuclear testing ground. Why not to provide TEC data from both the conjugate point and the same hemisphere site?

Response: We appreciate the reviewer for the valuable comment. In this work, in order to obtain smooth relative STEC data, only carries phase observation data of satellite elevation angle greater than 30° within 3 hours after the UNE are utilized to derive the relative STEC, which to some extent limit the number of observations. From Figure 5, we present the IPPs tracks of relative STEC derivatives. The red lines indicate the IPPs tracks obtained by IGS stations in the northern hemisphere. The blue lines indicate the magnetic conjugate positions of the IPPs tracks obtained by IGS stations in the southern hemisphere. It is found in the GPS dataset that there are no observation data (IPPs, ionospheric piecing points) in the vicinity of nuclear test site during the UNE event. Therefore, there is no way to investigate the response of TEC disturbance in the

[Figure]

vicinity of nuclear testing ground in this work.

Fig. 2. In this plot only the moment of TEC disturbance can be seen. However, the waveform of TEC disturbance is not shown anywhere. Additional Figure with extended time scale is needed.

Response: We appreciate the reviewer for the valuable comment. Figure 1 show the time sequences of raw data corresponding to relative STEC disturbances presented in Figure 4 in the revised manuscript. Compared with the magnitude and time scale of ionospheric disturbances caused by earthquakes presented in Kong et al. (2018), ionospheric disturbances presented in Figure 1 were relatively smaller and lasted with 5 minutes. It is difficult to found the ionospheric disturbances in response to UNE from the relative STEC time series. Therefore, the numerical third-order horizontal 3-point derivatives of relative STEC are used for extracting the ionospheric disturbances in this work.

Figure 1. The time sequences of raw data corresponding to relative STEC disturbances presented in Figure 4 in the revised manuscript. The ionospheric STEC disturbances in response to UNE are represented by the red rectangles.

Fig. 3. The same problem with this plot. Only the moment of FAC impulse can be seen, but not its waveform. Additional Figure with extended time scale is needed. Plot for another day is not necessary.

Response: We appreciate the reviewer for the valuable comment. Figure 2 show the time sequences of raw data corresponding to ionospheric current disturbances presented in Figure 6 in the revised manuscript. Compared with the magnitude of current disturbances in Figure 6, current disturbances presented in Figure 2 were relatively smaller. It is difficult to found the ionospheric disturbances in response to UNE from the current time series. Therefore, the numerical third-order horizontal 3-point derivatives of current are used for extracting the ionospheric disturbances in this work.

Figure 2. The time sequences of raw data corresponding to ionospheric current disturbances presented in Figure 6 in the revised manuscript. The ionospheric current disturbances in response to UNE are represented by the red rectangles.

Reference: Kong, J., Yao, Y., Zhou, C., Liu, Y., Zhai, C., Wang, Z., and Liu, L.: Tridimensional reconstruction of the Co-Seismic Ionospheric Disturbance around the time of 2015 Nepal earthquake, J. Geodesy, 3, 1-12, 2018.  Editorial comments: Fig. 1. Lines with geomagnetic coordinates are needed. The reference to Ren et al. (2012) is absolutely irrelevant. All the names in ref. at line 213 are misspelled. Few comments concerning interpretation: Theoretical model of FAC generation at the front of the acoustic pulse has been presented in [Pokhotelov O.A., Parrot M., Pilipenko V.A., Fedorov E.N., Surkov V.V., and Gladyshev V.A., Response of the ionosphere to natural and man-made acoustic sources, Annales Geophysicae, 13, N11, 1197- 1210, 1995; Pokhotelov O.A., Pilipenko V.A., Fedorov E.N., Stenflo L., and Shukla P.K., Induced electromagnetic turbulence in the ionosphere and the magnetosphere, Physica Scripta, 50, 600-605, 1994; Pokhotelov, O.A., Pilipenko V.A., and Parrot M., Strong atmospheric disturbances as a possible origin of inner zone particle diffusion, Annales Geophysicae, 17, 526-532, 1999].

Response: We appreciate the reviewer for the valuable comment. We have corrected accordingly. We would like to thank the reviewer again for the valuable comments, which help a lot to improve the quality of the present paper. We hope that the reviewers will be satisfied with our responses and revisions, and we look forward to hearing from the reviewers soon.

Please also note the supplement to this comment:
https://www.ann-geophys-discuss.net/angeo-2018-122/angeo-2018-122-AC4-supplement.zip

ANGEOD

Interactive
comment

[Figure]

**Fig. 1.** figure1

SUWN PRN 28

**Fig. 2.** figure2

**Fig. 3.** figure3

[Figure]

**Fig. 4.** figure4

**Fig. 5.** figure5

**Fig. 6.** figure6

**Fig. 7.** figure7

[Figure]

[Figure]

**Fig. 8.** figure8

**Fig. 9.** figure1_response3

[Figure]

Fig. 10. figure2_response3

---

## Author Response (AR2)

This manuscript reported an interesting event about ionospheric disturbances caused by the nuclear explosion which could be associated with LAIC electric field penetration rather than originating from AGW mechanism. This study is useful to understand the ionospheric response to the lithosphere events, including natural and human activities. However, I have some questions about conclusion and results. The authors should address these issues before the possible acceptation.

**Comments:**

1. It was mentioned that AGWs mechanism cannot explain the geomagnetic conjugate observation. However, the ionospheric disturbances caused by AGWs can also generated the electric field which can also map into the conjugate hemisphere. Jonah et al. (2017) suggested that the daytime MSTIDs (ionospheric disturbances) observed at conjugate hemispheres are caused by AGWs.

Jonah, O. F., E. A. Kherani, and E. R. De Paula (2017), Investigations of conjugate MSTIDs over the Brazilian sector during daytime, J. Geophys. Res. Space Physics, 122, 9576–9587, doi:10.1002/2017JA024365.

Response:

We appreciate the reviewer for the valuable comment. We totally agree with the reviewer that the ionospheric disturbances caused by AGWs could also generated the electric field which could also map into the conjugate hemisphere and further generate ionospheric disturbances in conjugate region. However, compared with TEC disturbances induced by MSTIDs presented in *Jonah et al.* (2017), ionospheric disturbances in response to North Korea UNE in both hemispheres were relatively smaller and lasted within 5 minutes in our work. Therefore, electric field disturbances induced by TEC disturbances presented in Figure 4 may be very small and cannot generate obvious ionospheric disturbances in conjugate region. Using the Thermosphere Ionosphere Electrodynamic General Circulation Model (TIEGCM), we simulate ionospheric effects associated with initial TEC disturbances with maximum value of 0.2 TECU in the region of the nuclear test site nearby during 03:30:00-03:40:00 UTC on 3 September, 2017. Figure 1 and Figure 2 present ionospheric effects at

03:40:00 UTC and 04:10:00 UTC, respectively. From Figure 1 and Figure 2, it was found that electric field disturbances with maximum value of 0.03 mV/m at around 350 km could be generated by TEC disturbances and map into the conjugate hemisphere. However, in our simulated results, there were no obvious TEC variations in conjugate region. The reason may be that the electric field variations in conjugate region were too small to cause the obvious TEC disturbances through $E \times B$ drift. Therefore, based on the simulated results, we propose that geomagnetic conjugate ionospheric disturbances in our work may be more likely to be caused by LAIC electric penetration rather than AGWs.

[Figure]

Figure 1. The global TEC variations associated with initial TEC disturbances in the

region of the nuclear test site nearby during 03:30:00-03:40:00 UTC on 3 September, 2017. The results for universal times of 03:40:00 UTC, and 04:10:00 UTC, respectively. The red hollow star indicates the location of the nuclear test site. The geomagnetic conjugate position of the nuclear test site is represented by blue hollow star.

[Figure]

Figure 2. The global electric field variations at around 350 km. The results for universal times of 03:40:00 UTC, and 04:10:00 UTC, respectively.

2. The schematic sketch of LAIC electric field was used to explain the concentric ionospheric disturbances. How could concentric ionospheric disturbances be caused by electric field?

Response:

We appreciate the reviewer for the valuable comment. Concentric ionospheric disturbance could be caused by concentric electric field through $\boldsymbol{E} \times \boldsymbol{B}$ drift. Using the

LAIC electric field penetration model proposed by *Zhou et al.* (2017), concentric electric field distribution at 90 km was shown in Figure 3. The while arrow indicates the direction of the total horizontal electric field at 90 km. Because of the existence of high conductivity of geomagnetic field, concentric electric field at the ionospheric bottom could map into the *F* region and conjugate hemisphere, resulting in concentric ionospheric disturbances by electrodynamic process through $E \times B$ drift.

[Figure]

Figure 3. The horizontal distribution of the ionospheric abnormal electric field generated by LAIC electric field coupling. The geographical latitude and longitude (41.35 °N, 129.11 °E).

3. The used method is useful to detect the ionospheric irregularities rather than ionospheric disturbances in this study. This could be the reason that the duration of ionospheric disturbances here is shorted than that observed by Kong et al. (2018).

Response:

We appreciate the reviewer for the valuable comment. TEC disturbances presented in

*Kong et al.* (2018) exceeded 0.3 TECU and lasted for 15-20 minutes during 2015 Nepal earthquake, while the UNE-generated ionospheric disturbances in our study were relatively smaller and lasted within 5 minutes. The reason for difference of TEC disturbances may be that earthquake magnitude and background ionosphere are different. We suggest that the numerical third-order horizontal 3-point derivatives of relative STEC could be used for extracting the ionospheric disturbances caused by natural earthquake or UNE events.

We would like to thank the reviewer again for the valuable comments, which help a lot to improve the quality of the present paper. We hope that the reviewers will be satisfied with our responses and revisions, and we look forward to hearing from the reviewers soon.

[revised manuscript text omitted]